# A Compact High-Order Finite-Difference Method with Optimized Coefficients for 2D Acoustic Wave Equation

**Liang Chen** [ID] **, Jianping Huang \*, Li-Yun Fu** [ID] **, Weiting Peng, Cheng Song and Jiale Han**

School of Earth Science, China University of Petroleum (East China), Qingdao 266000, China
* Correspondence: jphuang@upc.edu.cn

**Abstract:** High-precision finite difference (FD) wavefield simulation is one of the key steps for the successful implementation of full-waveform inversion and reverse time migration. Most explicit FD schemes for solving seismic wave equations are not compact, which leads to difficulty and low efficiency in boundary condition treatment. Firstly, we review a family of tridiagonal compact FD (CFD) schemes of various orders and derive the corresponding optimization schemes by minimizing the error between the true and numerical wavenumber. Then, the optimized CFD (OCFD) schemes and a second-order central FD scheme are used to approximate the spatial and temporal derivatives of the 2D acoustic wave equation, respectively. The accuracy curves display that the CFD schemes are superior to the central FD schemes of the same order, and the OCFD schemes outperform the CFD schemes in certain wavenumber ranges. The dispersion analysis and a homogeneous model test indicate that increasing the upper limit of the integral function helps to reduce the spatial error but is not conducive to ensuring temporal accuracy. Furthermore, we examine the accuracy of the OCFD schemes in the wavefield modeling of complex structures using a Marmousi model. The results demonstrate that the OCFD4 schemes are capable of providing a more accurate wavefield than the CFD4 scheme when the upper limit of the integral function is $0.5\pi$ and $0.75\pi$.

**Keywords:** wavefield simulation; compact finite difference; acoustic wave equation; numerical dispersion; optimization schemes

## 1. Introduction

Seismic forward modeling plays an important role in seismic data processing and interpretation and is the basis of migration and full-waveform inversion [1–5]. Acoustic wave equation has attracted wide interest in 2D and 3D seismic exploration due to its advantages, such as low computational costs and storage requirements. To date, finite difference (FD) methods [6–9], finite-element methods [10,11], spectral-element methods [12–15], boundary integral methods [16–19], pseudo-spectral methods [20–22], and finite volume methods [23] have been used to calculate the propagation of seismic waves in various media. Among them, FD methods are widely used in exploration seismology for their simple implementation and high efficiency.

Conventional FD methods can be simply classified into two categories: explicit and implicit (compact) schemes. Explicit FD schemes are based on recursion relations and are popular for numerically approximating wave equations due to their lower computational costs and simplicity [24,25]. However, the computational accuracy of lower-order explicit FD methods is lower, while the stability conditions of higher-order explicit FD schemes are more stringent [26,27]. The compact finite difference (CFD) schemes use a linear combination of function values to represent the derivative of the function, which can effectively improve accuracy and stability. Early in the 1970s, several compact Hermitian FD schemes were proposed to solve partial differential equations in the field of fluid mechanics [28–30]. In [31], the author introduced a family of CFD schemes with spectral-like resolution and went through a careful analysis of them. Mahesh [32] proposed a

more general version of the standard CFD schemes summarized by Lele [31]. Later, the compact spectral scheme was used to approximate first-order and second-order derivatives in the simulation of three-dimensional turbulent flows [33]. Combined CFD schemes were developed to solve the convection–diffusion equation [34], Navier–Stokes equation [35], and advection equation [36] because they are more accurate and compact than conventional CFD schemes. The CFD schemes mentioned above are implemented on regular grids. Geodheer and Potters [37] and Shukla and Zhong [38] reported the application of the CFD schemes on non-equidistant grids. Up to now, various CFD schemes have been widely used in aerodynamics, hydrodynamics, and electromagnetics.

The CFD schemes also show great potential in solving seismic wave equations in exploration seismology. Yang [39] applied the CFD schemes to model the elastic and acoustic wave propagation in a 2D TI medium. Du et al. [40] combined the CFD scheme with a staggered-grid technique, developed a compact staggered-grid FD method, and applied it to simulate elastic wave propagation in a VTI medium. Kosloff [41] proposed a general CFD scheme that can calculate the derivative value at the current point using the function values at any number of points and used it to solve the elastic and acoustic wave equation. Subsequently, Chu and Stoffa [42] and Liu [43] applied the CFD schemes to solve seismic wave propagation in the frequency domain. Liao [44] discussed the accuracy, stability, and dispersion of a Padé approximation-based CFD scheme for the numerical simulation of a 3D acoustic wave equation. In [45], the authors proposed a new CFD scheme to solve a 3D acoustic wave equation, which has fourth-order accuracy in time and space and is simple to implement.

Similar to explicit FD schemes, the CFD schemes also suffer from grid dispersion, especially when large time and spatial steps are involved. Increasing the order of a specific CFD scheme can suppress the dispersion to some extent. However, higher-order CFD schemes require more computing and storage costs. To decrease the dispersion error at a large wavenumber range, optimization-based strategies are usually adopted to obtain modified FD coefficients. Kim and Lee [46] applied the dispersion–relation–preserving (DRP) method proposed by Tam and Webb [47] to minimize the dispersive errors in the wavenumber domain and obtained the optimum coefficients of tridiagonal and pentadiagonal CFD schemes. Liu et al. [48] optimized the pentadiagonal compact scheme using a sequential quadratic programming method and demonstrated its increased performance. Yu [49] proposed an optimized DRP-combined CFD scheme to solve the advection equation. Based on the least-squares method, Venutelli [50] developed two optimized fourth-order CFD schemes and presented classical applications for 1D and 2D nonlinear shallow water equations.

In this paper, we will investigate the validity of a family of tridiagonal CFD schemes and its optimization schemes for solving the 2D acoustic wave equation in an isotropic medium. The rest of this paper is as follows. In Section 2, the new CFD schemes with optimized coefficients are derived, which is followed by the dispersion and stability analysis. In Section 3, two numerical examples are used to demonstrate the feasibility and effectiveness of the proposed method, which is followed by a discussion and a conclusion in Sections 4 and 5, respectively.

## 2. Theory and Methods

### 2.1. Tridiagonal Compact Finite-Difference Schemes for 2D Acoustic Wave Equation

Consider the 2D acoustic wave equation in an isotropic medium:

$$\frac{1}{v^2}\frac{\partial^2 u}{\partial t^2} = \frac{\partial^2 u}{\partial x^2} + \frac{\partial^2 u}{\partial z^2} + s(t) = \Delta u + s(t) \tag{1}$$

where $u(x, y, z)$ is the scalar acoustic wavefield, $v(x, y, z)$ is the velocity field, $s(t)$ denotes the source function, and $\Delta$ denotes the Laplace operator.

The key to solving Equation (1) is to approximate $\Delta u$ using high-order FD schemes. For simplicity, let $f(x)$ be a function of one variable, $x_i$ be the i-th grid point in the x-

direction, and $f_i$ be the value of $f(x_i)$. In [31], a multi-parameter family of pentadiagonal CFD schemes was proposed to approximate the second derivative of $f(x)$:

$$\beta f''_{i-2} + \alpha f''_{i-1} + f''_i + \alpha f''_{i+1} + \beta f''_{i+2} = \sum_{m=1}^{M} a_m \frac{f_{i+m} - 2f_i + f_{i-m}}{m^2 h_x^2} \tag{2}$$

where $M \in \{1, 2, 3\}$, $h_x$ denotes the grid interval in the x-direction, and $\alpha$, $\beta$ and $a_m$ are difference coefficients. By matching the Taylor series coefficients of different orders, we obtain the relations between $\alpha$ and $a_m$ as follows:

$$a_1 + a_2 + a_3 = 1 + 2\alpha + 2\beta \ \text{(second order)} \tag{3}$$

$$a_1 + 2^2 a_2 + 3^2 a_3 = \frac{4!}{2!}(\alpha + 2^2\beta) \ \text{(fourth order)} \tag{4}$$

$$a_1 + 2^4 a_2 + 3^4 a_3 = \frac{6!}{4!}(\alpha + 2^4\beta) \ \text{(sixth order)} \tag{5}$$

$$a_1 + 2^6 a_2 + 3^6 a_3 = \frac{8!}{6!}(\alpha + 2^6\beta) \ \text{(eighth order)} \tag{6}$$

$$a_1 + 2^8 a_2 + 3^8 a_3 = \frac{10!}{8!}(\alpha + 2^8\beta) \ \text{(tenth order)} \tag{7}$$

According to Equations (3)–(7), only the tenth-order compact scheme has unique FD coefficients since there are five coefficients to be determined. Other lower-order compact schemes are not uniquely determined because of insufficient constraint conditions.

For $\beta = 0$, a family of tridiagonal schemes is generated:

$$\alpha f''_{i-1} + f''_i + \alpha f''_{i+1} = \sum_{m=1}^{M} a_m \frac{f_{i+m} - 2f_i + f_{i-m}}{m^2 h_x^2} \tag{8}$$

when $M = 1$ and Equations (3) and (4) are chosen as constraints, the classical Padé scheme with fourth-order accuracy (CFD4) is recovered. If $M = 2$, we can obtain a sixth-order compact (CFD6) scheme constrained by Equations (3)–(5). For $M = 3$, an eighth-order compact (CFD8) scheme constrained by Equations (3)–(6) is obtained, which is the highest-order scheme that Equation (8) can achieve. The coefficients of these schemes are tabulated in Table 1. If a further choice of $\alpha = 0$ is made, conventional central FD schemes are obtained. We focus on the ability of Equation (8) to approximate $\Delta u$ in this paper.

**Table 1.** Coefficients of the tridiagonal compact schemes.

| Schemes | Constraints | Order | $\alpha$ | $a_1$ | $a_2$ | $a_3$ |
|---------|-------------|-------|----------|-------|-------|-------|
| CFD4 | Equations (3) and (4) | fourth | 0.1 | 1.2 | 0 | 0 |
| CFD6 | Equations (3)–(5) | sixth | 0.181818 | 1.090909 | 0.061818 | 0 |
| CFD8 | Equations (3)–(6) | eighth | 0.236842 | 0.967105 | 0.536842 | $-0.030263$ |

Equation (8) can be solved using the Thomas (chase) algorithm because the coefficient matrix consisting of $\alpha$ is tridiagonal. The computational effort of the Thomas algorithm for solving linear equations is (5X-4), where X is the number of grid points.

## 2.2. Optimization of Compact Finite-Difference Coefficients

By relaxing the constraints of each CFD scheme, we can obtain corresponding low-order schemes, which provides the possibility to improve the difference coefficients using various optimization methods. For example, if the CFD4 scheme is only restricted to Equation (3), it will degenerate into a new scheme with second-order accuracy.

Fourier analysis is usually used to measure the accuracy of the FD schemes. Let $w$ denote the product of the scaled true wavenumber $k$ and the sampling interval $h$. Applying

a Fourier transform to Equation (8), we obtain the scaled modified (numerical) wavenumber as follows:

$$L(w) = \sqrt{\frac{2\sum\limits_{m=1}^{M} a_m[1 - \cos(mw)]}{1 + 2\alpha\cos(w)}} \tag{9}$$

where $L(w)$ denotes the product of the scaled numerical wavenumber and the sampling interval $h$. We define a weighted deviation (integrated error) that aims to minimize $w$ and $L(w)$:

$$E(\alpha, a_1, a_2, a_3) = \int_0^t \left(L^2(w) - w^2\right)^2 G^2(w)dw \tag{10}$$

where $E$ denotes the error between $w$ and $L(w)$, $0 < t \leq \pi$ is the upper limit of the integral function, and $G(w)$ denotes the weighting function. The weighting function makes Equation (10) integrable analytically and reduces the numerical dispersion in the high-wavenumber ranges by weighting the integrated error. The following weighting function is adopted:

$$G(w) = 1 + 2\alpha\cos(w) \tag{11}$$

The optimization condition that makes $E$ a local minimum is as follows:

$$\frac{\partial E}{\partial \alpha} = 0 \tag{12}$$

Equations (3)–(6) and (12) provide a system of linear equations by which the optimization coefficients of the tridiagonal compact schemes can be determined. Table 2 shows the optimization schemes (OCFD) with different upper limits of integral functions.

**Table 2.** Coefficients of the tridiagonal compact optimization schemes.

| Schemes | Constraints | Order | Integral Limit | $\alpha$ | $a_1$ | $a_2$ | $a_3$ |
|---------|-------------|-------|---------------|----------|-------|-------|-------|
| OCFD4 | Equation (3) | second | $\pi$ | 0.166054 | 1.332109 | 0 | 0 |
| | | | $0.75\pi$ | 0.131511 | 1.263021 | 0 | 0 |
| | | | $0.5\pi$ | 0.112531 | 1.225063 | 0 | 0 |
| OCFD6 | Equations (3) and (4) | fourth | $\pi$ | 0.277327 | 0.963564 | 0.591090 | 0 |
| | | | $0.75\pi$ | 0.224304 | 1.034262 | 0.414346 | 0 |
| | | | $0.5\pi$ | 0.198053 | 1.069262 | 0.326844 | 0 |
| OCFD8 | Equations (3)–(5) | sixth | $\pi$ | 0.332545 | 0.751775 | 0.996214 | −0.082900 |
| | | | $0.75\pi$ | 0.277486 | 0.875656 | 0.731933 | −0.052617 |
| | | | $0.5\pi$ | 0.251903 | 0.933217 | 0.609136 | −0.038547 |

For comparison, as shown in Figure 1, we plot the difference error curves of these CFD and OCFD schemes in which the square of the numerical wavenumber varies with the true wavenumber. Figure 1a–c suggest that when the upper limit of the integral function is $0.5\pi$ or $0.75\pi$, the curves of the OCFD schemes are closer to the exact curve than that of the CFD scheme. When the upper limit of the integral function is $\pi$, over-optimization occurs in a certain wavenumber range. In addition, as the upper limit of the integral function decreases, the accuracy curve of the OCFD scheme gradually approaches that of the CFD scheme. In fact, the OCFD scheme will degenerate into the CFD scheme if the upper limit of the integral function infinitely approaches zero. Figure 1d displays a comparison of the CFD schemes and standard central FD schemes. We observe that the accuracy of the CFD schemes is much higher than that of the FD schemes of the same order.

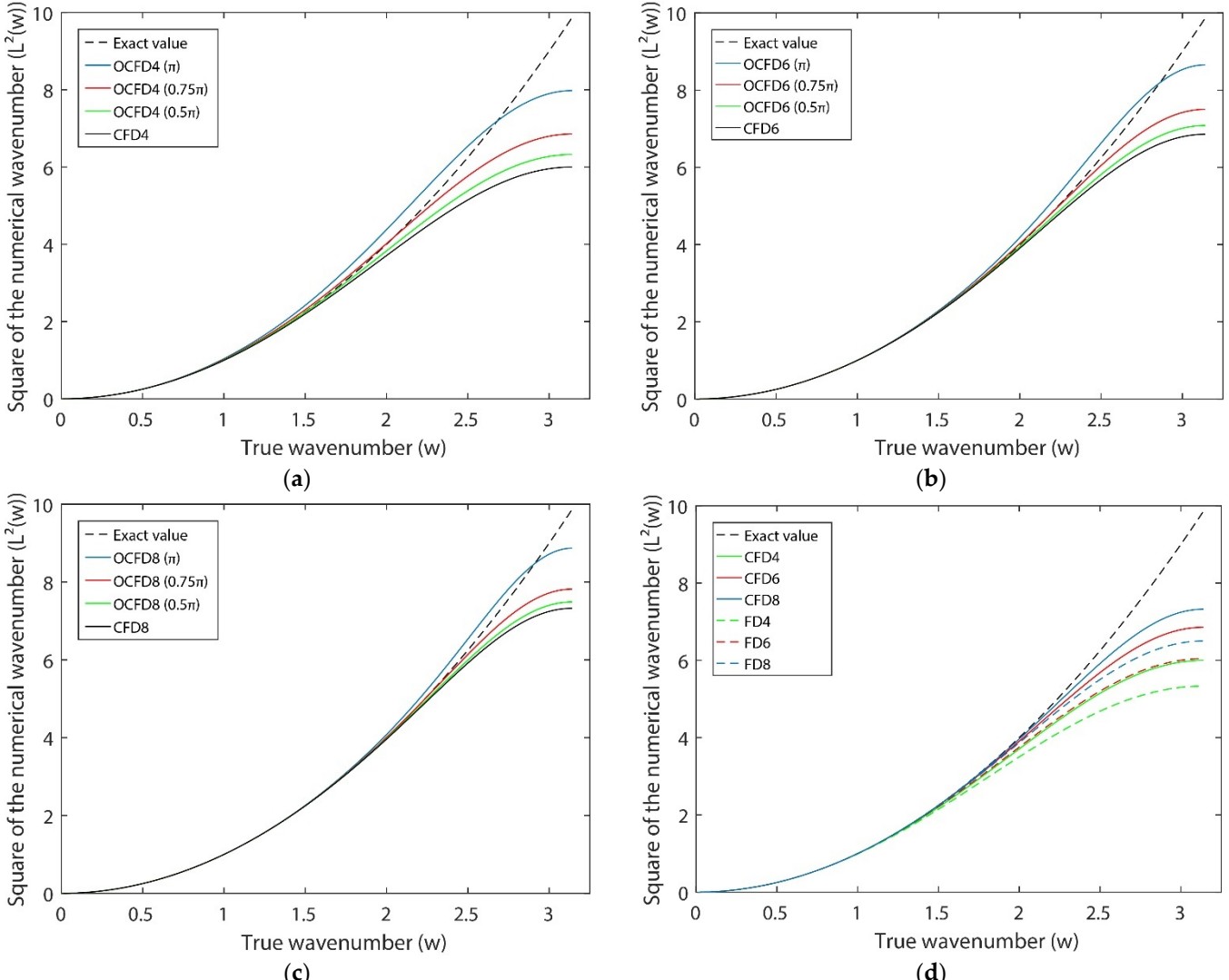

**Figure 1.** Accuracy curves: (**a**) comparison of the CFD4 and OCFD4 schemes, (**b**) comparison of the CFD6 and OCFD6 schemes, (**c**) comparison of the CFD8 and OCFD8 schemes, (**d**) comparison of the central FD and CFD schemes.

### 2.3. Dispersion Analysis and Stability Analysis

Numerical dispersion is a significant issue in FD forward modeling because the discretized form of the seismic wave equations is usually dispersive [51–53]. The dispersion can be measured by the error of the phase velocity $v_{FD}$ with respect to the true velocity $v_{TR}$. Let $u_{i,j}^n$ denote the numerical solution of the acoustic wavefield at time level $n\tau$ and grid point $(x_i, z_j)$. The numerical approximation of Equation (1) using second-order central finite-difference in time and various-orders compact finite-difference in space is written as:

$$u_{i,j}^{n+1} = 2u_{i,j}^n - u_{i,j}^{n-1} + \tau^2 v^2 \left[ (u_{xx})_{i,j}^n + (u_{zz})_{i,j}^n \right] \tag{13}$$

where $\tau$ denotes the time step. The spatial derivatives $u_{xx}$ and $u_{zz}$ are solved by Equation (8). Based on the plane-wave theory, we let:

$$u_{i,j}^n = \exp[I(ik_x h_x + jk_z h_z - \tilde{\omega} n\tau)] \tag{14}$$

where $\theta$ is the inclination angle between the $x$-axis and the direction of plane wave propagation, the vector wavenumber $\mathbf{k} = (k_x, k_z) = (|\mathbf{k}| \sin\theta, |\mathbf{k}| \cos\theta)$, $\widetilde{\omega} = v_{FD}|\mathbf{k}| = v_{FD}k$, and $I = \sqrt{-1} \in \mathbb{C}$.

Without loss of generality, let $h = h_x = h_z$. We obtain the dispersion relation of the CFD schemes by substituting Equation (14) into Equation (13):

$$\cos(\widetilde{\omega}\tau) = 1 - r^2 \left\{ \frac{\sum\limits_{m=1}^{M} a_m[1 - \cos(mhk_x)]}{2\alpha\cos(hk_x) + 1} + \frac{\sum\limits_{m=1}^{M} a_m[1 - \cos(mhk_z)]}{2\alpha\cos(hk_z) + 1} \right\} \tag{15}$$

where $r = \frac{v_{TR}\tau}{h}$. The phase velocity error is defined as follows:

$$\gamma = \frac{v_{FD}}{v_{TR}} - 1 = \frac{kv_{FD}\tau}{kv_{TR}\tau} - 1 = \frac{\widetilde{\omega}\tau}{k\frac{v_{TR}\tau}{h}h} - 1 = \frac{\widetilde{\omega}\tau}{krh} - 1 = \frac{\arccos(\cos(\widetilde{\omega}\tau))}{krh} - 1 \tag{16}$$

If $\gamma$ equals 0, the CFD scheme is non-dispersive. If $\gamma$ is negative and far from 0, strong spatial dispersion will occur. When $\gamma$ is positive, the temporal dispersion of the scheme is large. According to Equations (15) and (16), the phase error $\gamma$ depends on $\theta$ and $r$. Next, we fix one of them and calculate $\gamma$ for $kh$ ranging from 0 to $\pi$ to compare the dispersion curves for the other parameter.

First, we study the effect of $\theta$ on the dispersion. In Figure 2, we plot $\gamma$ vs. $kh$ for different $\theta$, while $r$ is fixed at 0.1. From Figure 2, when $r$ and $\theta$ are fixed, the phase velocity error of the higher-order schemes is much smaller than that of the lower-order schemes. From Figure 2a–f or 2g–i, we observe that the spatial dispersion of these schemes decreases as $\theta$ increases in the high wavenumber ranges. In addition, the OCFD schemes show great potential in eliminating the spatial dispersion in the high wavenumber ranges as the upper limit of the integral function increases. However, they are not suitable for suppressing the temporal dispersion, especially when the upper limit of the integral function is $\pi$. When the upper limit of the integral function is $0.5\pi$, we obtain three relatively good optimization schemes: OCFD4 ($0.5\pi$), OCFD6 ($0.5\pi$), and OCFD8 ($0.5\pi$). The temporal error of these optimization schemes is smaller than that of other optimization schemes.

Next, we fix the other parameter at $\theta = 0$ to study the effect of $r$. Figures 2a,d,g and 3 show that all the CFD and OCFD schemes are sensitive to $r$, which indicates that using a small time step may help to reduce the temporal dispersion error when the spatial grid interval is fixed.

Stability analysis is the other key issue in measuring the effect of an FD scheme. Almost all of the FD methods for approximating the seismic wave equations are conditionally stable and subject to constraints on velocity, spatial grid interval, and time step. The stability condition of the CFD schemes mentioned above can be derived by the Fourier method:

$$v_{\max}\tau\sqrt{\frac{1}{h_x^2} + \frac{1}{h_z^2}} \leq \sqrt{\frac{2(1 - 2\alpha)}{\sum\limits_{m=1}^{M} a_m\left[1 - (-1)^m\right]}} = C \tag{17}$$

where $v_{\max}$ denotes the maximum velocity of the model, and $C$ is the Courant–Friedrichs–Lewy (CFL) number. Table 3 shows the CFL number of the CFD and OCFD schemes. We observe that the CFL number of the lower-order CFD schemes is larger than that of the higher-order schemes. It should also be noted that the stability condition of the OCFD schemes is slightly stricter than that of the general scheme.

**Table 3.** CFL number of the CFD and OCFD schemes.

| Schemes | CFL Number | Schemes | CFL Number | Schemes | CFL Number |
|---|---|---|---|---|---|
| CFD4 | 0.817 | CFD6 | 0.764 | CFD8 | 0.750 |
| OCFD4 (0.5$\pi$) | 0.795 | OCFD6 (0.5$\pi$) | 0.752 | OCFD8 (0.5$\pi$) | 0.745 |
| OCFD4 (0.75$\pi$) | 0.764 | OCFD6 (0.75$\pi$) | 0.730 | OCFD8 (0.75$\pi$) | 0.735 |
| OCFD4 ($\pi$) | 0.708 | OCFD6 ($\pi$) | 0.680 | OCFD8 ($\pi$) | 0.708 |

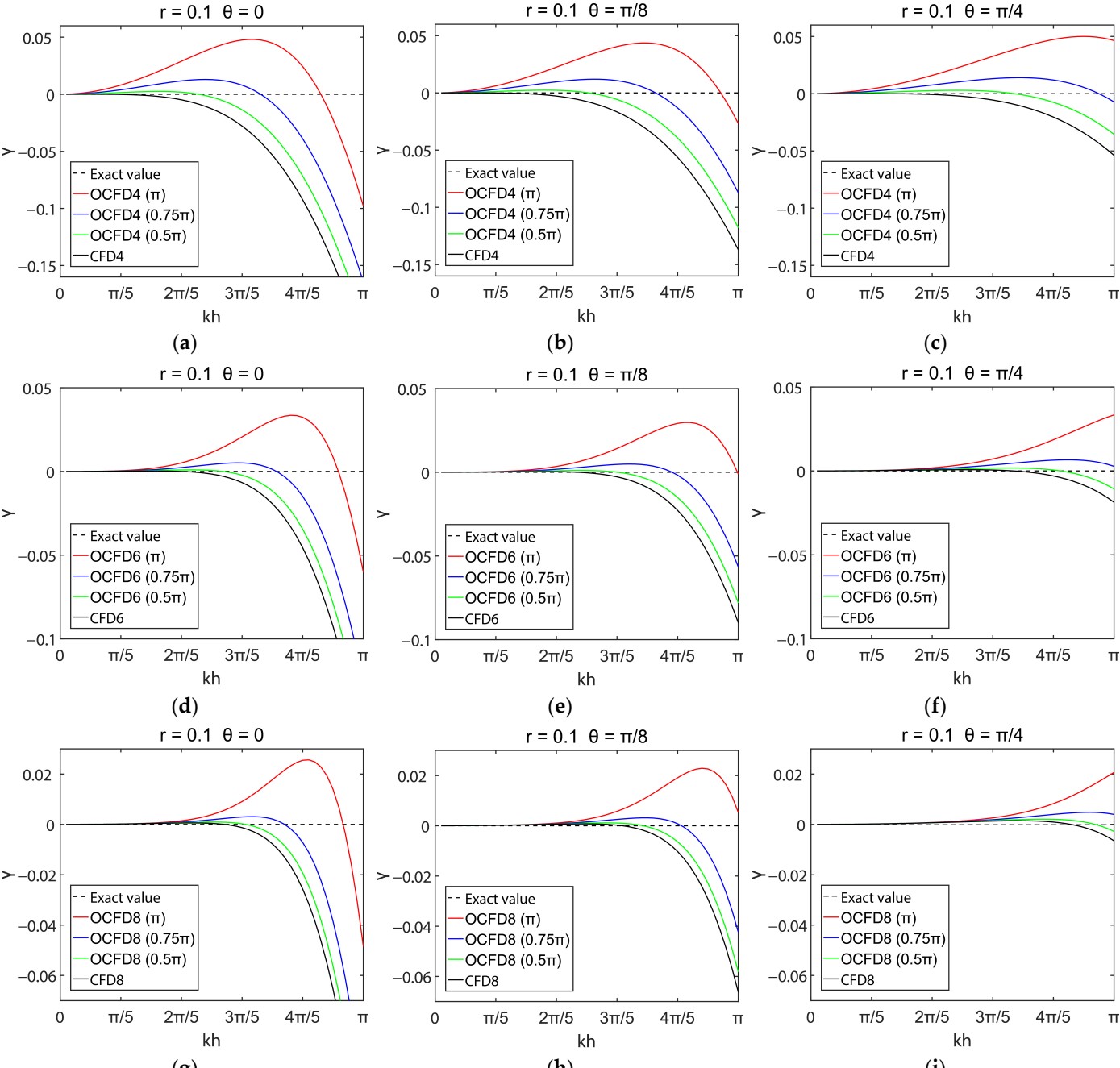

**Figure 2.** Dispersion curves of various $\theta$: (**a**–**c**) the CFD4 scheme and its optimization schemes, (**d**–**f**) the CFD6 scheme and its optimization schemes, (**g**–**i**) the CFD8 scheme and its optimization schemes.

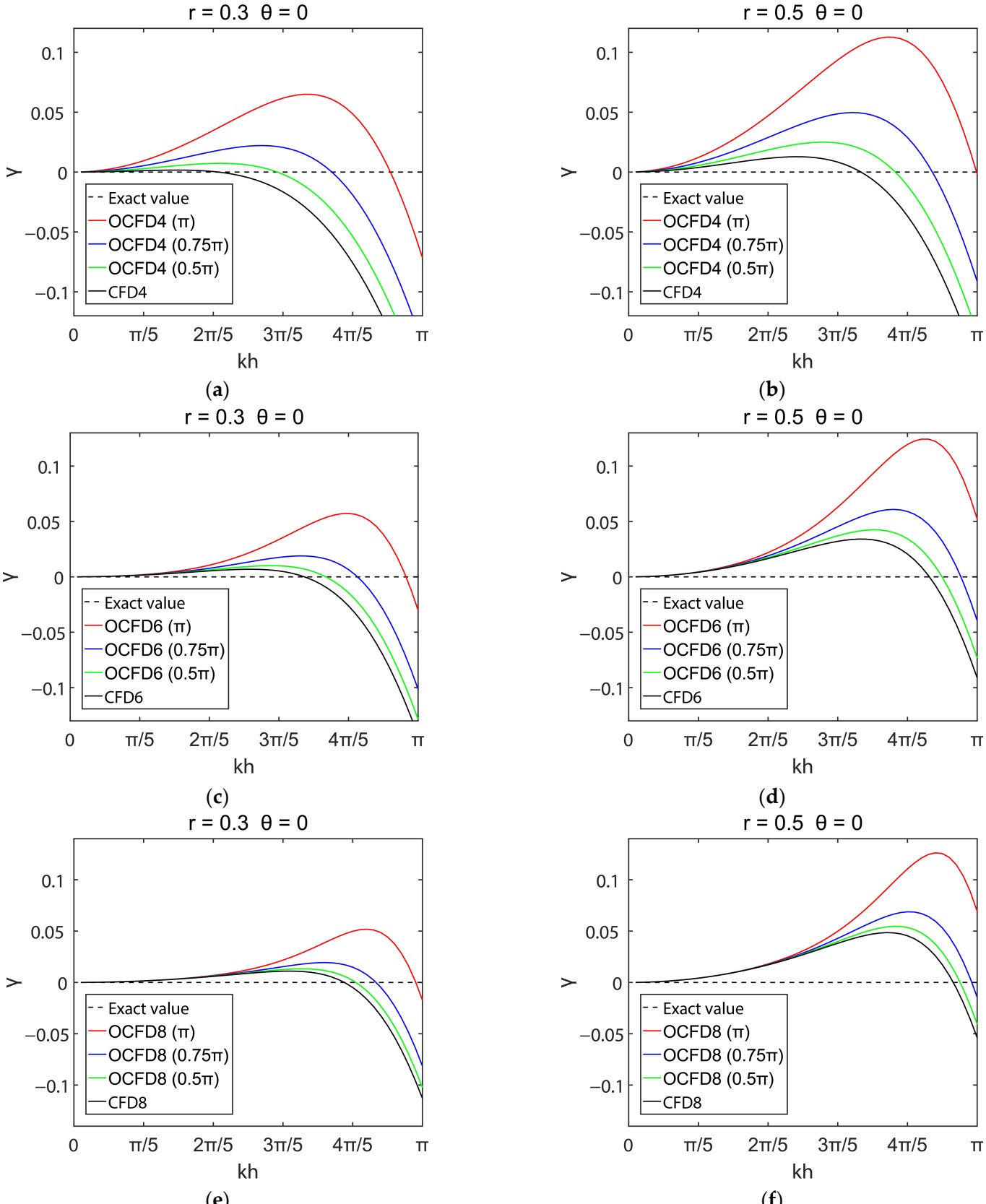

**Figure 3.** Dispersion curves of various *r*. (**a**,**b**) the CFD4 scheme and its optimization schemes, (**c**,**d**) the CFD6 scheme and its optimization schemes, (**e**,**f**) the CFD8 scheme and its optimization schemes.

## 3. Numerical Examples

### 3.1. Homogeneous Model

We use a simple homogeneous model to evaluate the simulation accuracy of the new schemes shown in Equation (13). The model is discretized into $200 \times 200$ grids. The velocity of this model is 3000 m/s. A Ricker wavelet with 30 Hz dominant frequency is used for the simulation, which is located at the center of the model. The time step is 1 ms and the spatial grid interval is 20 m. Figure 4 displays the snapshots at 0.5 s simulated by the CFD4 scheme (Figure 4a), OCFD4 schemes (Figure 4b–d), and conventional central FD schemes (Figure 4e,f). In Figure 4a, the spatial grid dispersion is strong because the spatial grid interval is large. In Figure 4b–d, the spatial dispersion is gradually eliminated as the upper limit of the integral function increases (indicated by the red arrows). However, when the upper limit of the integral function is π, the time error generated in acoustic wave propagation is too large to be ignored (indicated by the black arrow).

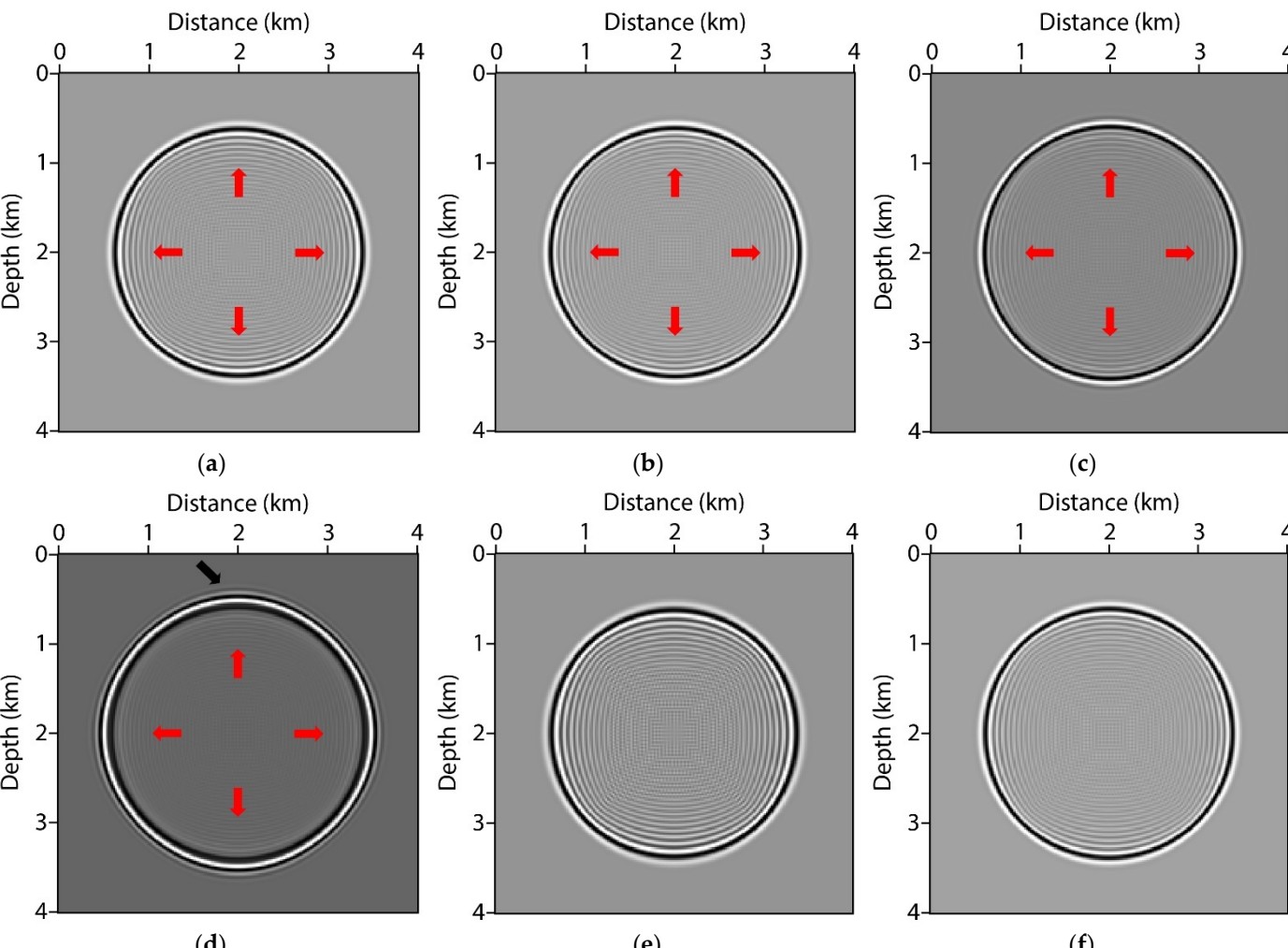

**Figure 4.** Snapshots of the homogeneous model at 0.5 s. They are simulated by: (**a**) CFD4 scheme, (**b**) OCFD4 scheme (0.5π), (**c**) OCFD4 scheme (0.75π), (**d**) OCFD4 scheme (π), (**e**) fourth-order central FD scheme, and (**f**) sixth-order central FD scheme, respectively. The red arrows indicate the spatial dispersion and the black arrow indicates the temporal dispersion.

We extract four traces from the snapshots simulated by the CFD4 and OCFD schemes for further analysis, as shown in Figure 5a. The dashed arrows indicate the spatial numerical dispersion, and the dashed box indicates the temporal dispersion. The single trace waveform of the CFD4 and OCFD4 (π) schemes between 0.4 and 0.8 km is clearly different

from that of other optimization schemes, which shows that increasing the upper limit of the integral function contributes less to temporal simulation accuracy. To prove that the simulation accuracy of the proposed scheme is better than that of conventional central FD schemes, we extract single traces at a lateral distance of 2 km from the snapshots shown in Figure 4b,e,f. The result is shown in Figure 5b. From Figure 5b, we conclude that the modeling accuracy of the OCFD4 (0.5π) scheme is not inferior to that of the sixth-order central FD scheme.

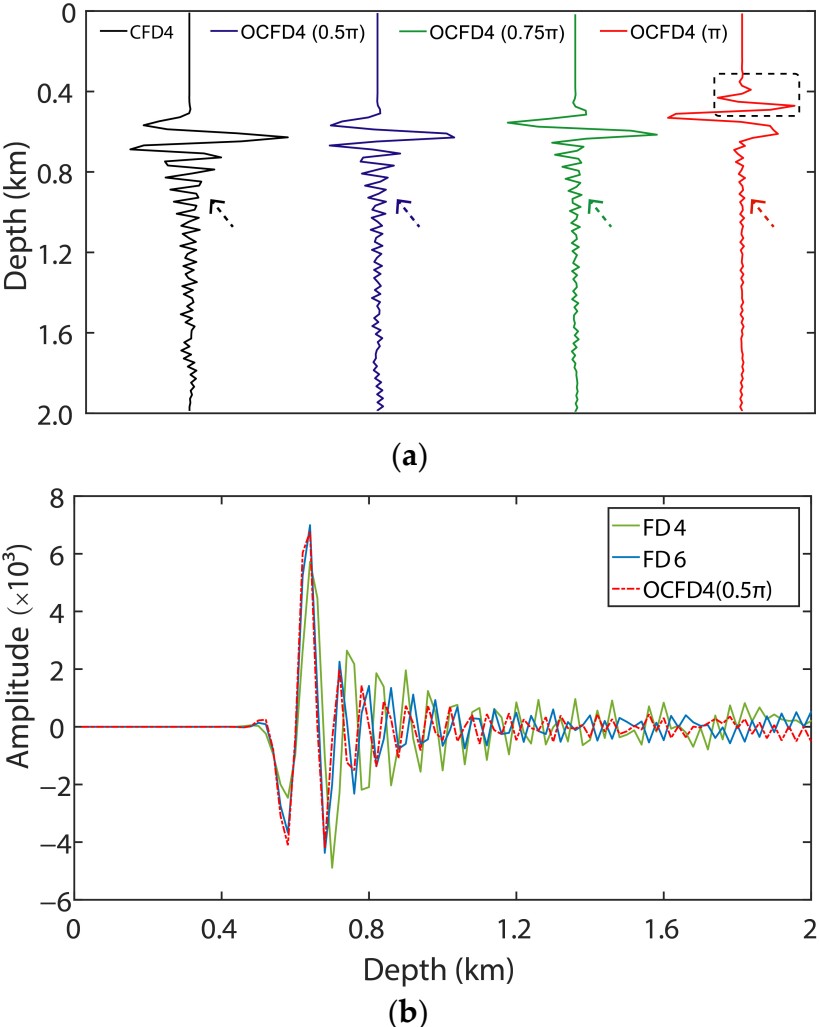

**Figure 5.** Single traces of the snapshots at lateral distance of 2 km. (**a**) Comparison of the CFD4 and OCFD4 schemes and (**b**) comparison of the OCFD4 (0.5π) and central FD schemes. The dashed box indicates the temporal dispersion.

The snapshots simulated by the CFD6 and CFD8 schemes and their optimization schemes are shown in Appendix A. The simulation results displayed in Figures A1 and A2 are similar to those shown in Figure 4. Table 4 displays the computing time for the CFD schemes of different orders. The result shows that the computational cost of the CFD4 scheme is much lower than that of the higher-order schemes.

**Table 4.** Computing time for the CFD schemes of different orders.

| Schemes | Recording Time (s) | Computing Time (s) | Recording Time (s) | Computing Time (s) | Recording Time (s) | Computing Time (s) |
|---|---|---|---|---|---|---|
| CFD4 | 5 | 36.7 | 10 | 73.2 | 20 | 145.6 |
| CFD6 | 5 | 70.9 | 10 | 142.0 | 20 | 283.7 |
| CFD8 | 5 | 108.5 | 10 | 217.4 | 20 | 433.1 |

*3.2. Marmousi Model*

We use a Marmousi model to further illustrate the accuracy and effectiveness of the OCFD4 schemes in modeling acoustic wave propagation. The model is shown in Figure 6, which is discretized with $460 \times 250$ grids. The spatial grid spacing is 10 m and the time step is 0.5 ms. The source is located at (2300, 10) m, which is a Ricker wavelet with 30 Hz dominant frequency. A perfectly matched layer absorbing boundary [54,55] is used for wavefield extrapolation.

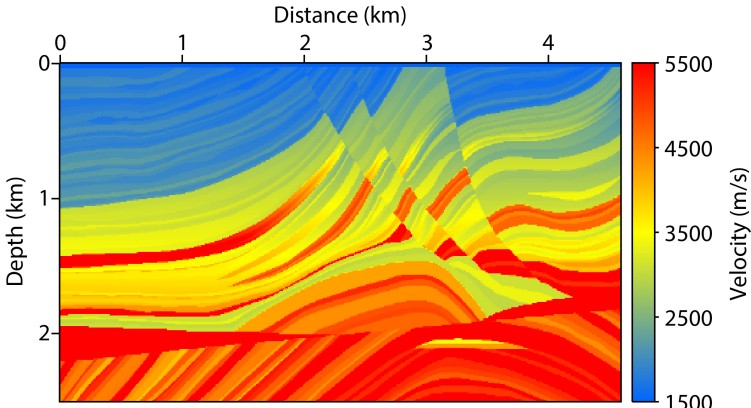

**Figure 6.** Marmousi model.

Figure 7 displays the wavefield snapshots at 0.9 s computed by the CFD4 (Figure 7a) and OCFD4 (Figure 7c–e) schemes. The snapshot in Figure 7b is simulated by the CFD6 scheme, which is used as a reference. When the upper limit of the integral function is $0.5\pi$ or $0.75\pi$, the wavefield is calculated correctly. As indicated by the red arrows in Figure 7e, the wavefield modeled by the OCFD4 ($\pi$) scheme is different from those simulated by other schemes. Figure 8 shows the difference in the simulated wavefield. The error of the residual wavefield in Figure 8a–d relative to the reference wavefield in Figure 7b at L2-norm is 1.65%, 1.06%, 1.25%, and 1.74%, respectively. Obviously, the OCFD4 ($0.5\pi$) scheme is the most helpful in improving the simulation accuracy of the Marmousi model. Figure 9 displays the shot records simulated by the CFD4, CFD6, and OCFD4 ($\pi$) schemes, respectively. We extract two traces from the records for further comparison, as shown in Figure 10. The result shows that the simulation accuracy of the OCFD4 ($\pi$) scheme is better than that of the CFD4 scheme. From Figures 7–10, we conclude that the OCFD4 ($\pi$) scheme is effective, and it can be used to obtain more accurate simulation results than the general CFD4 scheme.

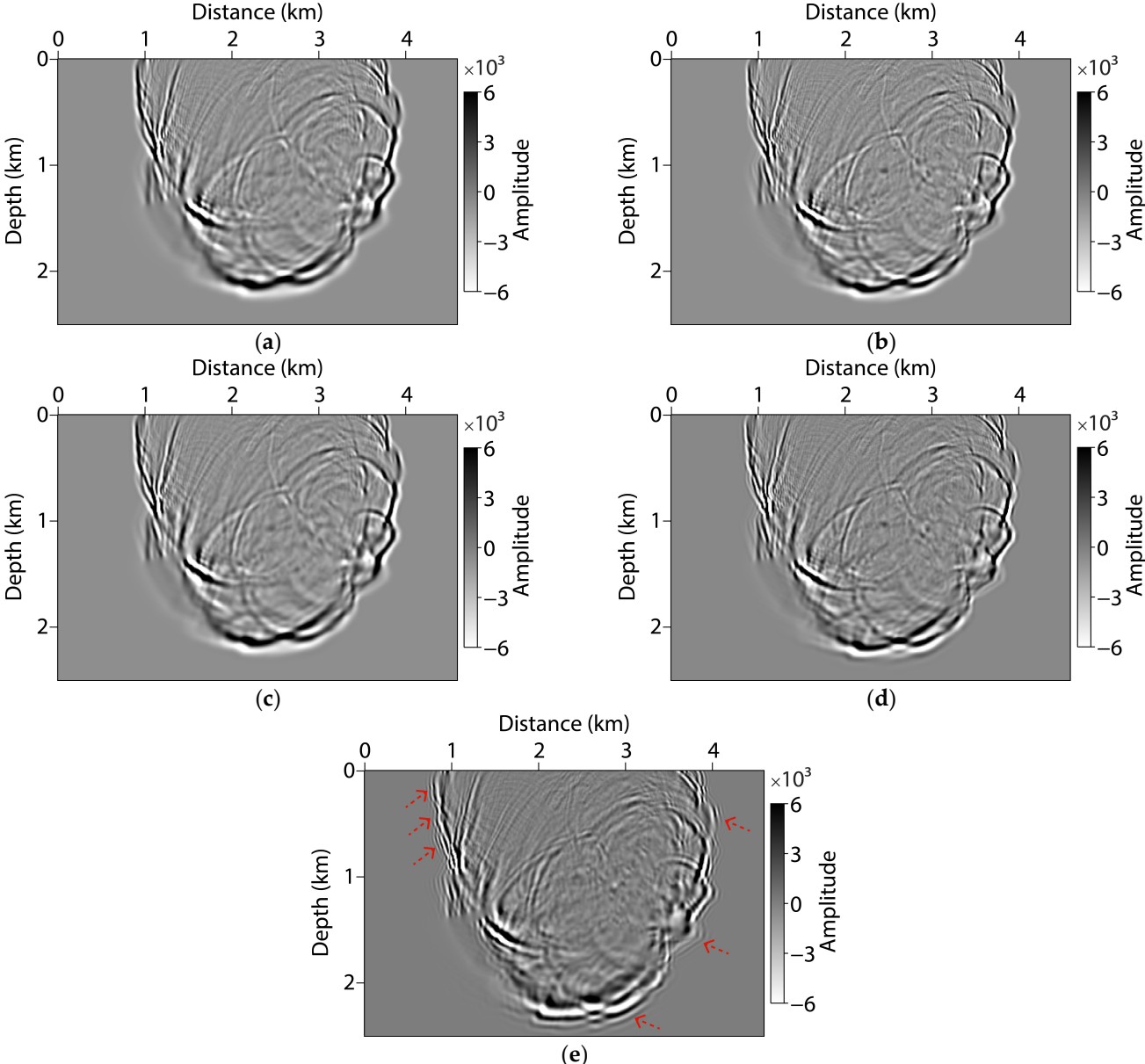

**Figure 7.** Snapshots of the Marmousi model at 0.5 s. They are simulated by: (**a**) CFD4 scheme, (**b**) CFD6 scheme, (**c**) OCFD4 scheme (0.5$\pi$), (**d**) OCFD4 scheme (0.75$\pi$), and (**e**) OCFD4 scheme ($\pi$), respectively. The red arrows indicate the dispersion error.

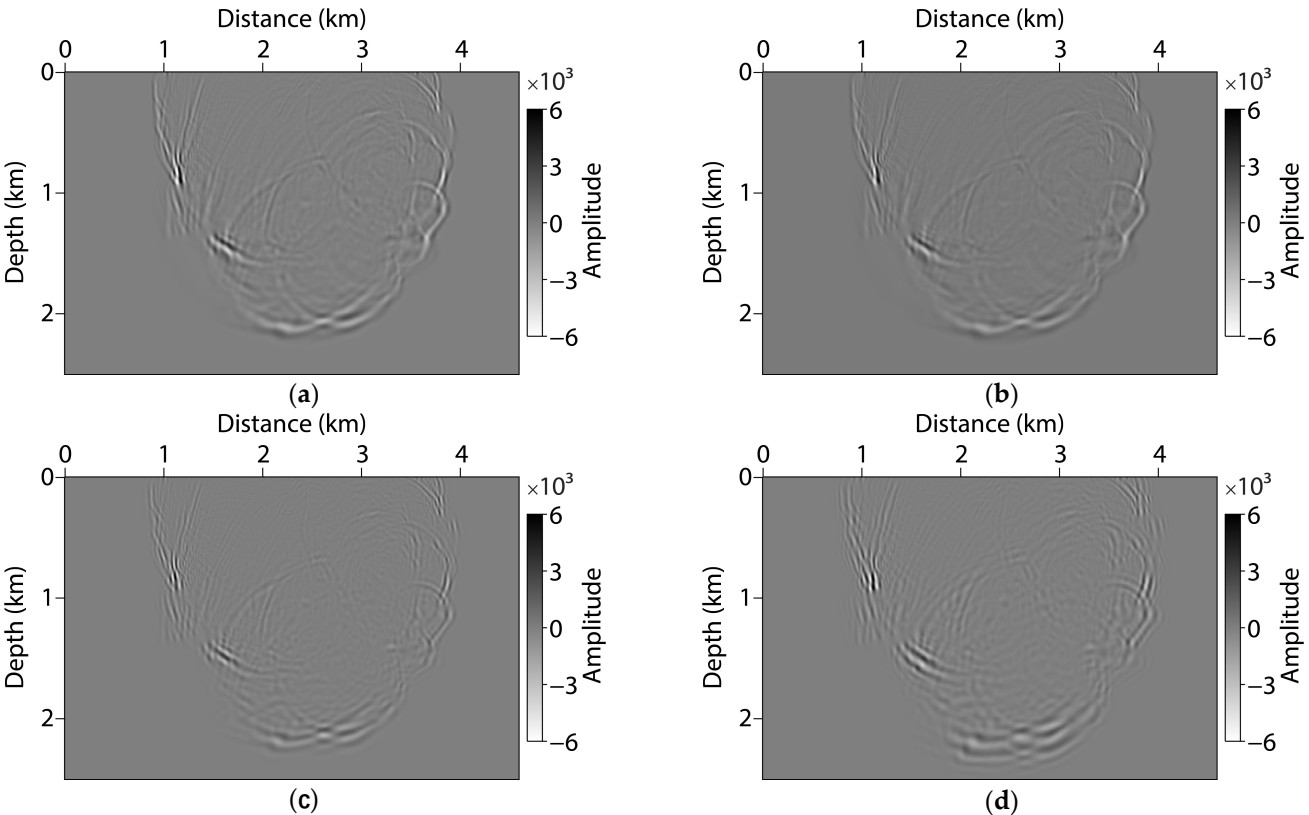

**Figure 8.** The difference of the simulated wavefield between: (**a**) Figure 7a,b, (**b**) Figure 7b,c, (**c**) Figure 7b,d, and (**d**) Figure 7b,e.

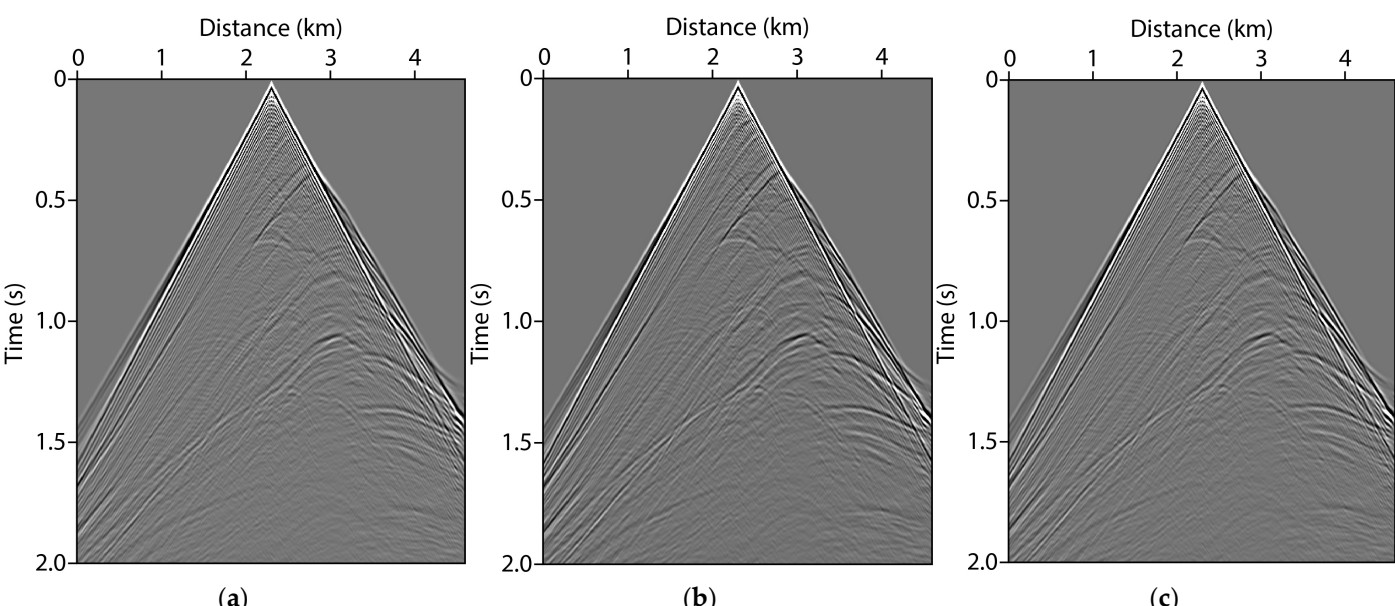

**Figure 9.** Shot records simulated by the: (**a**) CFD4 scheme, (**b**) CFD6 scheme, (**c**) OCFD4 (0.5π).

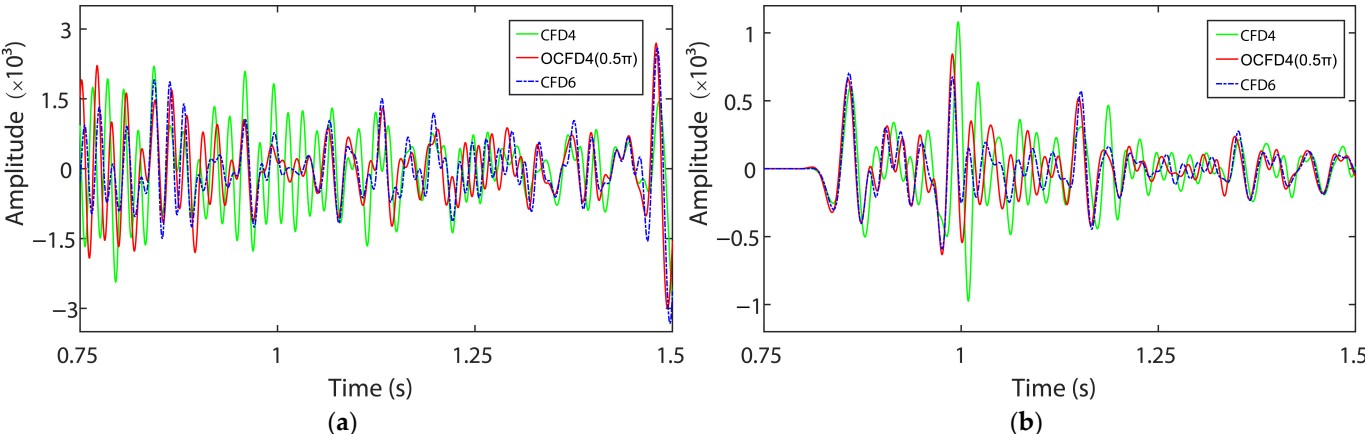

**Figure 10.** Single traces exacted from the shot records in Figure 9 at the distance of (**a**) 1.5 km and (**b**) 3.75 km.

## 4. Discussion

Compared to CFD schemes, most explicit FD schemes often used in seismic forward modeling are not compact, which gives rise to difficulty in dealing with boundary conditions. For example, a sixth-order central FD scheme requires a 13-points stencil and three layers of boundary conditions for approximating the second derivative in 2D cases while the CFD4 scheme with similar approximation accuracy only requires a 5-point stencil and one layer of boundary conditions. In 3D cases, the CFD4 scheme requires a 7-points stencil and one layer of boundary conditions, while the sixth-order central FD scheme requires a 19-point stencil and three layers of boundary conditions. Therefore, the CFD schemes require fewer boundary conditions than the explicit FD schemes. This advantage of the CFD schemes also helps to save some storage space in numerical simulation.

We studied three CFD schemes and their corresponding optimization schemes in this paper. In fact, the other two CFD schemes have more than one optimization scheme except for the CFD4 scheme. For example, if the CFD6 scheme is only constrained by Equation (3), a second-order OCFD6 scheme is generated. In this case, two optimization conditions, such as Equation (12), must be solved to make Equation (10) a local minimum. For the second-order OCFD8 scheme, it is necessary to solve a system of equations consisting of three optimization conditions to obtain the optimization coefficients. All possible tridiagonal optimization schemes are summarized in Table A1 of Appendix B. The coefficients for the OCFD6_2, OCFD8_2, and OCFD8_4 schemes are not given in the paper due to the limitation of computational power. In addition, Equation (10) can be regarded as a least square problem. Therefore, some optimization algorithms, such as the minimax approximation method [56], Newton method [57], and conjugate gradient method [58], can be used to obtain these coefficients.

The selection of the upper limit of the integral function has a significant impact on the OCFD schemes. For each optimization scheme, we only discussed three representative cases where the upper limit of the integral is $0.5\pi$, $0.75\pi$, and $\pi$, respectively. The coefficients in these cases are easily obtained by solving Equation (12) because some terms cancel out in the calculation. The weighting function is also important for optimization results. Some exponential weighting functions [46,59] can help to obtain better optimization coefficients by weighting the integrated error in the high wave number range. However, it makes it difficult to find the primitive function of Equation (10).

## 5. Conclusions

We have proposed a high-accuracy forward modeling scheme for simulating acoustic wave propagation. A family of tridiagonal OCFD schemes with 2M-order accuracy in space and the central FD scheme with second-order accuracy in time are used to approximate the spatial and temporal second derivatives, respectively. The accuracy curves show that the

CFD schemes are superior to conventional FD schemes of the same order, and the OCFD schemes are better than the CFD schemes in a certain wavenumber range. The dispersion analysis demonstrates that increasing the upper limit of the integral function helps to reduce spatial error but contributes less to improving temporal dispersion. In addition, the wavefield simulations on a homogeneous model confirm the conclusion drawn in the dispersion analysis. The efficiency tests for the homogeneous model show that the lower-order CFD scheme is more effective than the higher-order schemes. The numerical experiments on the Marmousi model suggest that the OCFD4 ($0.5\pi$ and $0.75\pi$) schemes facilitate improvement in the wavefield modeling of complex structures.

**Author Contributions:** Conceptualization, L.C. and J.H. (Jianping Huang); formal analysis, W.P.; investigation, C.S. and J.H. (Jiale Han); methodology, L.C.; resources, J.H. (Jianping Huang) and L.-Y.F.; software, C.S. and J.H. (Jiale Han); supervision, J.H. (Jianping Huang); writing—original draft, L.C.; writing—review and editing, J.H. (Jianping Huang). All authors have read and agreed to the published version of the manuscript.

**Funding:** This research is supported by the Marine S&T Fund of Shandong Province for Pilot National Laboratory for Marine Science and Technology (Qingdao) (grant number 2021QNLM020001), the National Key R&D Program of China (grant number 2019YFC0605503C), the National Outstanding Youth Science Foundation (grant number 41922028), and the Major Scientific and Technological Projects of China National Petroleum Corporation (grant number ZD2019-183-003).

**Data Availability Statement:** Not applicable.

**Conflicts of Interest:** The authors declare no conflict of interest.

**Appendix A**

Figure A1 shows the snapshots of the homogeneous model simulated by the CFD6 and its optimization schemes. Figure A2 shows the snapshots of the homogeneous model simulated by the CFD8 and its optimization schemes. The two figures are shown below:

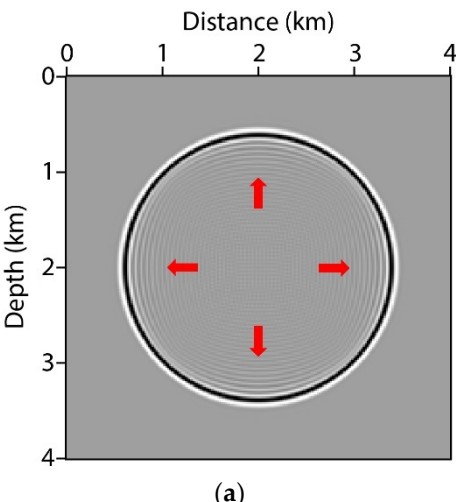

(a)

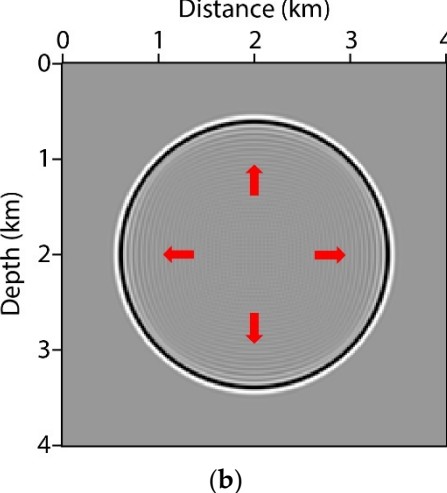

(b)

**Figure A1.** *Cont.*

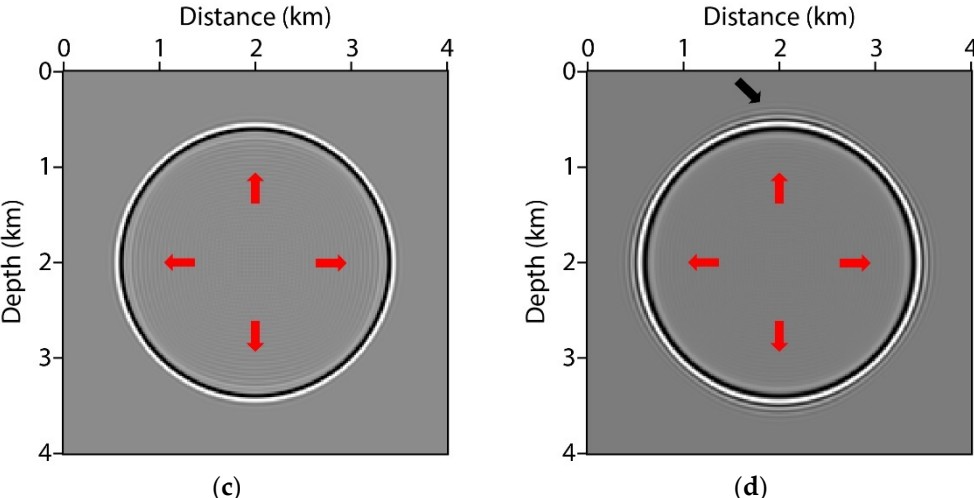

**Figure A1.** Snapshots of the homogeneous model at 0.5 s. They are simulated by: (**a**) CFD6 scheme, (**b**) OCFD6 scheme ($0.5\pi$), (**c**) OCFD6 scheme ($0.75\pi$), and (**d**) OCFD6 scheme ($\pi$), respectively. The red arrows indicate the spatial dispersion and the black arrow indicates the temporal dispersion.

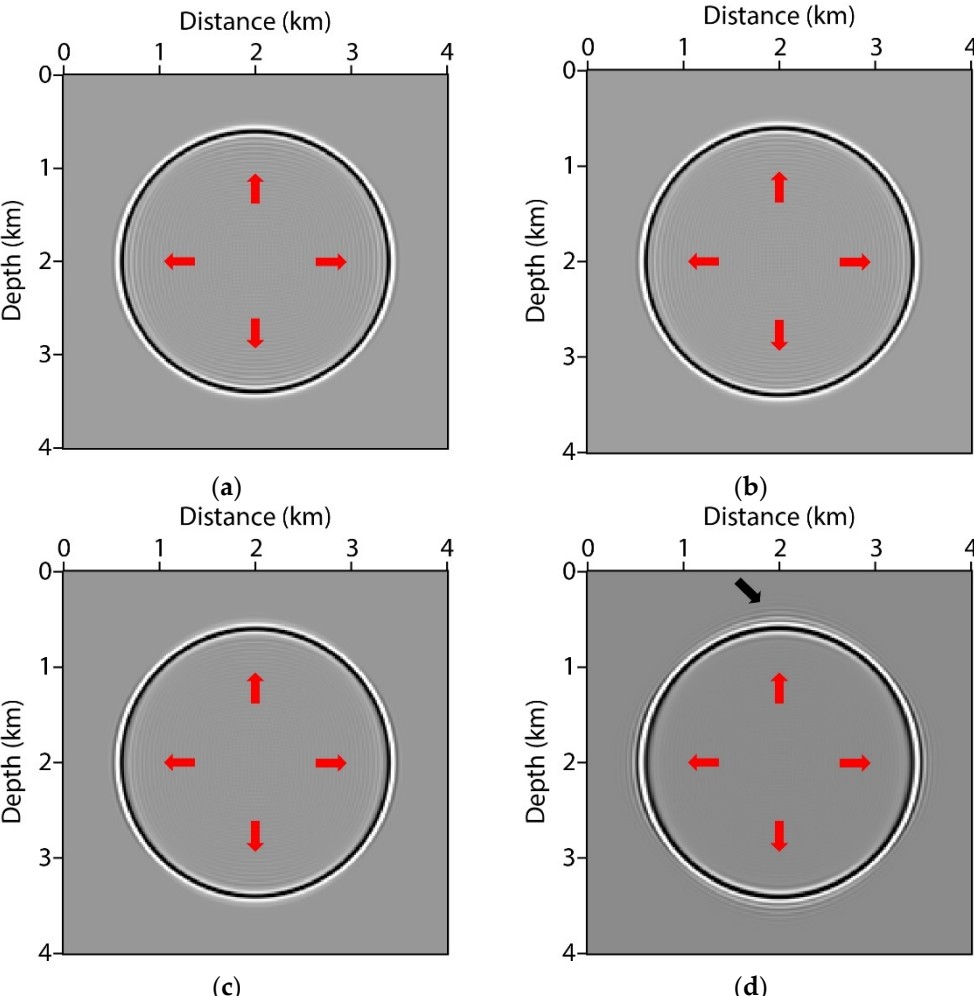

**Figure A2.** Snapshots of the homogeneous model at 0.5 s. They are simulated by: (**a**) CFD8 scheme, (**b**) OCFD8 scheme ($0.5\pi$), (**c**) OCFD8 scheme ($0.75\pi$), and (**d**) OCFD8 scheme ($\pi$), respectively. The red arrows indicate spatial dispersion and the black arrow indicates the temporal dispersion.

**Appendix B**

Table A1 shows all tridiagonal optimization schemes with different orders. OCFD6_2 denotes the second-order OCFD6 scheme, OCFD8_4 denotes the fourth-order OCFD8 scheme, and OCFD8_2 denotes the second-order OCFD8 scheme.

**Table A1.** All tridiagonal optimization schemes with different orders.

| Schemes | Constraints | Order | Optimization Conditions |
|---------|-------------|-------|-------------------------|
| CFD4 | Equations (3) and (4) | fourth | / |
| OCFD4 | Equation (3) | second | $\partial E / \partial \alpha = 0$ |
| CFD6 | Equations (3)–(5) | sixth | / |
| OCFD6 | Equations (3) and (4) | fourth | $\partial E / \partial \alpha = 0$ |
| OCFD6_2 | Equation (3) | second | $\partial E / \partial \alpha = 0, \partial E / \partial a_1 = 0$ |
| CFD8 | Equations (3)–(6) | eighth | / |
| OCFD8 | Equations (3)–(5) | sixth | $\partial E / \partial \alpha = 0$ |
| OCFD8_4 | Equations (3) and (4) | fourth | $\partial E / \partial \alpha = 0, \partial E / \partial a_1 = 0$ |
| OCFD8_2 | Equation (3) | second | $\partial E / \partial \alpha = 0, \partial E / \partial a_1 = 0, \partial E / \partial a_2 = 0$ |

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
