# Peer review of "A Compact High-Order Finite-Difference Method with Optimized Coefficients for 2D Acoustic Wave Equation"

_remotesensing, doi:10.3390/rs15030604_

Round 1
Reviewer 1 Report
Dear Editor,
All my comments are in the attached pdf file.
Cheers,
Gang Yao

Author Response
Dear reviewer:
We feel great thanks for your professional review work on our article. As you are concerned, there are several problems that need to be addressed. According to your nice suggestions, we have made extensive corrections to our previous draft, the detailed corrections are listed below. To make the reply more visible, Q represents questions and A are our answers for these questions.
Please see the attachment for more information.

Reviewer 2 Report
Chen et al. proposed a high-precision forward modeling scheme for simulating acoustic wave propagation, and derived the corresponding optimization scheme by minimizing the error between the real wavenumber and the numerical wavenumber. Then, an optimized CFD (OCFD) scheme and a second-order central FD scheme are used to approximate the spatial and temporal derivatives of the two-dimensional acoustic wave equation. The accuracy curves show that the CFD scheme is better than the central FD scheme of the same order, and the OCFD scheme is better than the CFD scheme in a specific wavenumber range. This method can effectively solve the problems that the explicit FD format of the seismic wave equation is not compact enough, and the processing of boundary conditions is difficult and inefficient. However, in the text, the definition of some parameters is not clear. Although the authors have tested the stability and dispersion, I have some reservations about the explanation in the text. It is suggested that this article needs to be revised before it can be published in this journal.
Comments:
Lines 110-111: The authors mentioned that lower-order compact schemes are not uniquely determined because of insufficient constraint conditions, what are the constraint conditions? Please give some examples.
Lines 128-129: The w was defined as the product of the scaled true wavenumber k and the sampling interval h, but in Line 134, the w is the true wave number, and also in Line 178, w= VFD k. From the above definitions that this manuscript marked, the definition of w is seemed some different, please unify the statement in the text.
Lines 150-151: It is still unclear that why the upper limit of integral function is 0.5 pi or 0.75 pi, the curves of the OCFD schemes are closer to the exact curve than that of the CFD scheme. Whether these two values have special meaning in the formula which is not seen in this manuscript, please verify it clearly. Although the authors mention that the coefficients in these cases are easily obtained by solving equation (12), since some terms are canceled out in the calculation.
Lines 178-179: Please define what is VFD, and there seems to be something missing in the formula at the end of this sentence.
Line 183: Please define what is VTR. It seems not be mentioned in the manuscript.
Lines 202-203: The authors mentioned that all the CFD and OCFD schemes are sensitive to r, and imply that a small time step may help to reduce the temporal dispersion error when the spatial grid interval is fixed, under this assumption, what are the high frequencies that the CFD and OCFD schemes can resolve?
Lines 309-312: The authors mentioned that the advantage of the CFD approach helps to save some storage space, especially in 3D, but I think it is difficult just use requires a 5-points stencil and one layer of boundary condition. Maybe the authors can test some cases in 3D to verify the arguments.
Author Response
Dear reviewer:
On behalf of all the contributing authors, I would like to express our sincere appreciations of your letter and constructive comments concerning our article. These comments are all valuable and helpful for improving our article. According to your comments, we have made extensive modifications to our manuscript and supplemented extra data to make our results convincing. Point-by-point responses are listed below this letter. To make the reply more visible, Q represents questions and A are our answers for these questions.
Please see the attachment for more information.

Reviewer 3 Report
The manuscript proposes a new algorithm for computing the coefficients of the finite-difference numerical scheme for the two-dimensional acoustic equation. The proposed method does not require significant changes in existing numerical schemes, while increasing the accuracy of the solution without increasing computational costs. The accuracy and stability of the proposed scheme was evaluated theoretically and numerically.
I would recommend to better disclose the significance of the research for remote sensing.
There are minor typographical errors in the text and layout.
I think that this manuscript deserves publication in the remote sensing journal after minor corrections.
Author Response
Dear reviewer:
Thank you for your decision and constructive comments on our manuscript. According to your nice suggestions, we have made extensive corrections to our previous draft.
Round 2
Reviewer 1 Report
The author has revised it according to the preliminary review opinions. The article meets the publishing requirements and can be published.
Author Response
Dear reviewer
We feel great thanks for your professional review work on our article. We wish you all the best in your work.
Reviewer 2 Report
Most of the questions in this article have been correctedThere are also some unclear arguments that need to be clarified. A minor revision to improve this article is necessary to publish in this scientific research originated journal.
Lines 129-136: Althrough the authors explain the “W” and “ w ” represent different physical meanings, But for readers, it is easy to confuse these two parameters, and their symbols are very similar. I suggest that authors can replace w(omega) with w~. Or the parameter can replace it with another symbol.
,
Author Response
Dear reviewer,
We feel great thanks for your professional review work on our article. According to your nice suggestion, we have changed w (omega) to w~ in Eqs 14-16. Please see Lines 179 for more details.